# AHA! A̲NIMATING H̲UMAN A̲VATARS IN DIVERSE SCENES WITH GAUSSIAN SPLATTING

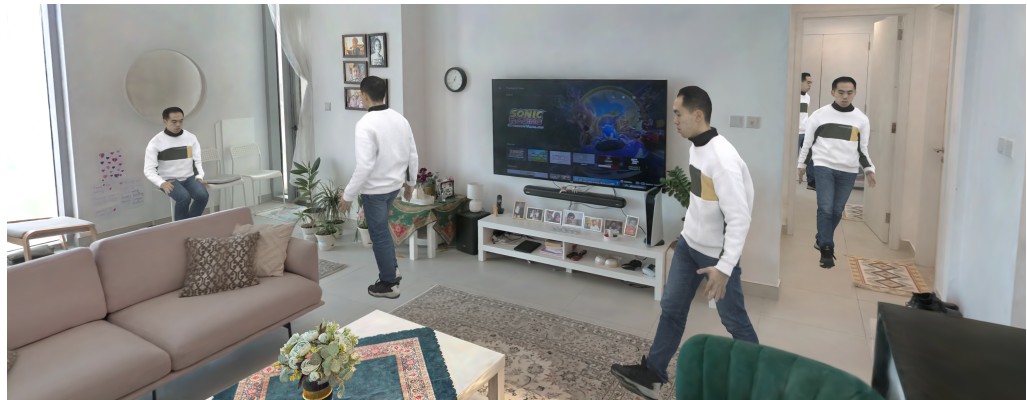

Figure 1. We extend 3D Gaussian splatting to human animation, introducing a unified Gaussian-based representation for both humans and environments. This enables dynamic synthesis of human–scene interactions and photorealistic rendering with the Gaussian splatting algorithm, demonstrating a new direction for neural scene representations in animation.

## ABSTRACT

We present a novel framework for animating humans in 3D scenes using 3D Gaussian Splatting (3DGS), a neural scene representation that has recently achieved state-of-the-art photorealistic results for novel-view synthesis but remains underexplored for human-scene animation and interaction. Unlike existing animation pipelines that use meshes or point clouds as the underlying 3D representation, our approach introduces the use of 3DGS as the 3D representation to the problem of animating humans in scenes. By representing humans and scenes as Gaussians, our approach allows for geometry-consistent free-viewpoint rendering of humans interacting with 3D scenes. Our key insight is that the rendering can be decoupled from the motion synthesis and each sub-problem can be addressed independently, without the need for paired human-scene data. Central to our method is a Gaussian-aligned motion module that synthesizes motion without explicit scene geometry, using opacity-based cues and projected Gaussian structures to guide human placement and pose alignment. To ensure natural interactions, we further propose a human–scene Gaussian refinement optimization that enforces realistic contact and navigation. We evaluate our approach on scenes from Scannet++ and the SuperSplat library, and on avatars reconstructed from sparse and dense multi-view human capture. Finally, we demonstrate that our framework allows for novel applications such as geometry-consistent free-viewpoint rendering of edited monocular RGB videos with new animated humans, showcasing the unique advantage of 3DGS for monocular video-based human animation.

## 1 INTRODUCTION

Human animation in 3D scenes is essential for applications ranging from video gaming and computer-generated imagery (CGI) to robotics. Recent research has made significant progress on generating humans in 3D scenes (Hassan et al., 2019; Jiang et al., 2024a; Hassan et al., 2021b; Zhao et al., 2023;

Hwang et al., 2025), where humans are typically represented as either 3D skeletons or meshes, while background scenes are represented as meshes or point clouds. These representations are compact and versatile, capable of modeling a wide variety of surfaces. However, a fundamental limitation persists: there is almost always a domain gap between rendered results and real images due to limitations in lighting, materials, and geometric fidelity.

In parallel, neural scene representations have emerged, beginning with NeRF (Mildenhall et al., 2020) and recently 3D Gaussian Splatting (Kerbl et al., 2023), enabling photorealistic rendering of objects, humans, and full 3D scenes from novel viewpoints or in novel poses. Yet, despite their success in rendering quality, neural representations have seen little to no adoption in human-scene animation pipelines, which continue to rely on mesh and point cloud–based frameworks.

Gaussian Splatting as a 3D representation for human-scene animation, *in theory* offers natural advantages over existing mesh based representations: First, 3DGS enables photorealistic rendering of human-scene interactions with superior lighting and material fidelity. Second, 3DGS allows for reconstructing scenes with only a monocular video (Ling et al., 2024) captured from a mobile phone, thus allowing for applications such as geometry consistent free viewpoint rendering of videos with new humans, personalized content creation and gaming in scenes from mobile-captured videos. Such applications are difficult with a mesh based representation as estimating meshes, or pointclouds from only monocular scene videos remains challenging (Wang et al., 2025). This motivates the central question addressed in this paper: *Can neural scene representations—specifically Gaussian Splatting—be effectively used as a 3D representation for human animation in 3D scenes?* (Fig. 1).

Several obstacles prevent a direct extension of 3DGS to human animation in 3D scenes. First, most existing work on human–scene interaction synthesis (Hassan et al., 2021b; Jiang et al., 2024a; Hwang et al., 2025; Zhao et al., 2023) assumes paired human motion data with scene geometry. Such datasets are difficult to collect at scale, and conversion from mesh-based annotations into Gaussians is non-trivial. Second, human-scene animation requires motion synthesis that respects both the structure of the scene and the natural dynamics of the human pose, which for a non-mesh representation remains non-trivial. Furthermore, unlike meshes, 3DGS does not provide explicit topology or clean geometry, complicating tasks like surface-based contact modeling.

To address these challenges, we offer a novel perspective for human–scene animation, grounded in two key insights: First, rendering of humans and scenes in 3DGS can be decoupled from motion synthesis. That is, we can reconstruct humans and scenes independently, animate humans in a canonical space, and then place them back into reconstructed 3DGS scenes. This is common in classical graphics pipelines for meshes, where canonical models are animated via skinning or rigging, and has recently been adopted for animatable 3DGS avatars as well. However, prior work has primarily studied such Gaussian avatars in isolation. In contrast, our contribution is to extend this paradigm to human–scene animation, where avatars must not only be animated but also consistently placed and rendered inside reconstructed 3DGS scenes. Second, motion synthesis can itself be decoupled from explicit geometry: even though 3DGS does not provide watertight surfaces, we show that its opacity fields and projected Gaussian structures offer sufficient cues to guide human placement.

Our framework proceeds in two stages. First, we reconstruct humans as animatable Gaussians from either multiview capture using an off-the-shelf module. These Gaussians are then posed using **a Gaussian-aligned motion module** (Sec. 3.2), which computes scene-aligned motion parameters by relying on opacity-based culling and orthographic projection of scene Gaussians for path finding. Our core contribution is the migration and adaptation of traditional scene-mesh human interaction techniques (including RL-based locomotion and motion transitions) to 3DGS. To further ensure realistic interactions, we introduce a human–scene **Gaussian refinement optimization** (Sec. 3.3) that adjusts the placement and motion of humans for natural contact and navigation within the scene.

To showcase the applicability of our method on diverse datasets, we present results on scenes from the Scannet++ dataset (Yeshwanth et al., 2023) and from scenes downloaded from the publicly available SuperSplat 3DGS library (SuperSplat). To demonstrate the efficacy of our method on different Avatar reconstruction datasets, we also showcase results on Avatars from the BEHAVE (Bhatnagar et al., 2022) and Avatarrex datasets (Zheng et al., 2023). The results from BEHAVE demonstrate that our method works on avatars reconstructed from sparse camera setups. We finally demonstrate the utility of our presented framework for geometry consistent free viewpoint rendering of monocular videos

with new animated humans on several monocular videos from the DL3DV dataset (Ling et al., 2024), showcasing the unique advantage of 3DGS for casual video-based human animation.

To summarize our contributions are as follows:

- We introduce the 3D Gaussian Splatting representation to the classical Computer Graphics problem of animating humans in 3D environments

- We demonstrate that our framework can be used for geometry consistent free viewpoint rendering of monocular videos edited with new animated humans

- We introduce a novel Gaussian aligned motion module for motion synthesis in scenes represented as 3D Gaussians

- We introduce a human scene Gaussian refinement optimization for correct placement of human Gaussians in scenes represented using 3DGS leading to better contact and interactions.

## 2 RELATED WORK

**Neural Rendering** Following the publication of NeRF (Mildenhall et al., 2020), there has been significant research on Neural Rendering (Xie et al., 2022b). Nerf is limited by its computational complexity and despite several follow-up improvements (Müller et al., 2022; Barron et al., 2022; 2023; Tancik et al., 2023), the high computational cost of NeRF rermain. 3DGS introduced in (Kerbl et al., 2023) addresses this limitation by representing scenes with an explicit set of primitives shaped as 3D Gaussians, extending previous work (Lassner & Zollhöfer, 2021). 3DGS rasterizes Gaussian primitives into images using a splatting algorithm (Westover, 1992). 3DGS originally designed for static scenes has been extended to dynamic scenes (Shaw et al., 2023; Luiten et al., 2024; Wu et al., 2024; Lee et al., 2024; Li et al., 2023a), slam-based reconstruction, (Keetha et al., 2024), mesh reconstruction (Huang et al., 2024; Guédon & Lepetit, 2024) and NVS from sparse cameras (Mihajlovic et al., 2024).

**Human Reconstruction and Neural Rendering** Mesh-based templates (Pavlakos et al., 2019; Loper et al., 2015) have been used to recover 3D human shape and pose from images and video (Bogo et al., 2016; Kanazawa et al., 2018). However, this does not allow for photoreal renderings. In (Alldieck et al.; 2019) recover a re-posable human avatar from monocular RGB. However their use of a mesh template also does not allow for photorealistic renderings. Implicit functions (Mescheder et al., 2019; Park et al., 2019) have also been utilized to reconstruct detailed 3D clothed humans (Chen et al., 2021; Alldieck et al., 2021; Saito et al., 2020; He et al., 2021; Huang et al., 2020; Deng et al., 2020). However, they are also unable to generate photorealistic renderings and are often not reposable. Several works (Peng et al., 2021; Guo et al., 2023; Weng et al., 2022; Jiang et al., 2022; Habermann et al., 2023; Zhu et al., 2024; Li et al., 2022; Liu et al., 2021; Xu et al., 2021) build a controllable NeRF that produces photorealistic images of humans from input videos. Unlike us, they do not model human-scene interactions. With the advent of 3DGS, several recent papers use a 3DGS formulation (Kocabas et al., 2023; Qian et al., 2024; Moreau et al., 2024; Abdal et al., 2024; Zielonka et al., 2023; Moon et al., 2024; Li et al., 2024b; Pang et al., 2024; Lei et al., 2023; Hu et al., 2024; Li et al., 2024a; Zheng et al., 2024; Jiang et al., 2024b; Dhamo et al., 2024; Qian et al., 2023; Xu et al., 2024; Junkawitsch et al., 2025) to build controllable human or face avatars. Unlike our method, they do not model human-scene interactions. Prior works have also extended the 3DGS formulation to model humans along with their environment, (Xue et al., 2024; Zhan et al., 2024; Mir et al., 2025), but unlike us, they either do not focus on human animation in 3D scenes.

**Humans and Scenes** Human-scene interaction is a recurrent topic of study in computer vision and graphics. Early works (Fouhey et al., 2014; Wang et al., 2017; Gupta et al., 2011) model affordances and human-object interactions using monocular RGB. The collection of several recent human-scene interaction datasets (Hassan et al., 2021a; Guzov et al., 2021; Hassan et al., 2019; Savva et al., 2016; Taheri et al., 2020; Bhatnagar et al., 2022; Jiang et al., 2024a; Cheng et al., 2023; Zhang et al., 2022) has allowed the computer vision community to make significant progress in joint 3D reconstruction of human-object interactions (Xie et al., 2022a; 2023; 2024a; Zhang et al., 2020). These datasets have also led to the development of methods that synthesize object conditioned controllable human motion (Zhang et al., 2022; Starke et al., 2019b; Hassan et al., 2021c; Diller & Dai, 2024). All these methods represent humans and scenes as 3D meshes and inherit the limitations of mesh-based representations

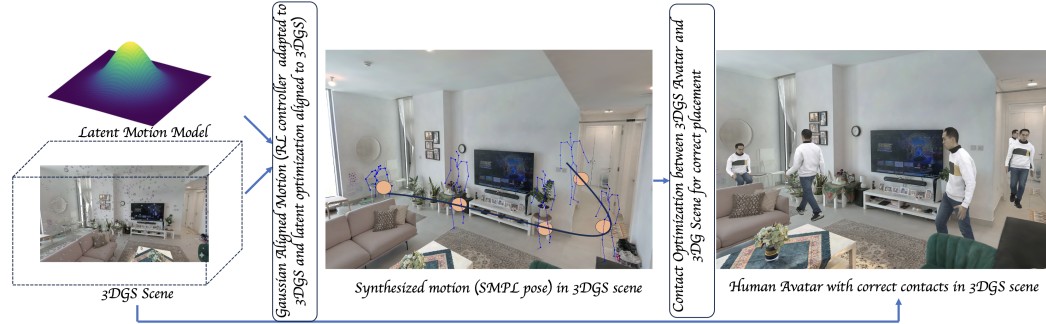

Figure 2: Using a latent motion model and 3DGS scene representation, we synthesize human motion that confirms with the 3D scene Gaussians using *Gaussian-aligned motion module*. We adapt RL based controllers and latent optimization for 3DGS scenes. We further refine these Gaussians for correct placements and contact. These composited human and scene Gausssians can be rendered from any viewpoint to generate photoeal images.

including their inability to generate photorealistic images, while our method allows for photorealistic renderings of humans and scenes.

**Human Animation** Human animation is another extensively studied problem in vision and graphics. Motion matching (Reitsma & Pollard, 2007), learned motion matching (Clavet, 2016; Holden et al., 2020) and motion graphs (Lee et al., 2002; Fang & Pollard, 2003; Kovar et al., 2008; Safonova et al., 2004; Safonova & Hodgins, 2007) are common methods employed in the video-gaming industry for generating kinematic motion sequences. Deep learning variants (Holden et al., 2017; Starke et al., 2019a; 2021; 2020) have also gained popularity. Diffusion Models (Tevet et al., 2023) have emerged as a powerful paradigm for human motion synthesis. Several follow-up works extend the original Motion Diffusion model with physics(Yuan et al., 2023), blended-positional encoding (Barquero et al., 2024), and for fine-grained controllable motion synthesis (Karunratanakul et al., 2023; Pinyoanuntapong et al., 2024; Xie et al., 2024b). Reinforcement learning (Zhang & Tang, 2022; Zhao et al., 2023) is another oft-used paradigm used for motion synthesis. Diffusion models have also been used as latent-motion models (Zhao et al., 2025) but unlike us, they only focus on clean, noise-free mesh-based scene representations and ignore the rendering aspects of human-scene interaction leading to limited rendering quality and photorealism.

## 3 METHOD

We present a method that enables virtual humans represented using Gaussian splats to navigate and interact in complex environments reconstructed as 3D Gaussian scenes. Our framework consists of three key components: **(1) Gaussian Reconstruction:** We reconstruct both scenes and humans as 3D Gaussians from RGB images. For scenes, we use standard 3DGS reconstruction, while for humans, we learn *human Gaussian* representations that can be animated with different SMPL poses (Sec. 3.1). **(2) Gaussian-Aligned Motion Synthesis:** Central to our approach is a novel *Gaussian-aligned motion module* (Sec. 3.2), which uses the reconstructed scenes (Sec. 3.1) and a latent-variable based motion synthesis framework (using RL and latent space optimization adapted to 3DGS) to synthesize motion parameters aligned with the 3DGS scenes **(3) Differentiable Contact Refinement in 3DGS:** We use the synthesized human motion data to animate human Gaussians and apply a novel refinement algorithm for correct human-scene interaction (Sec. 3.3) (Fig. 3). The refinement module detects contact frames from motion data and optimizes translation vectors to enforce proper contact between human and scene Gaussians while maintaining temporal smoothness and avoiding penetrations. These composed Gaussians can be rendered from any camera viewpoint to produce videos of humans interacting with diverse scenes. Figure 2 provides an overview.

### 3.1 GAUSSIAN RECONSTRUCTION

**Scene Gaussians.** Given monocular or multi-view video of a static scene, we model the environment as a set of $N_S$ anisotropic 3D Gaussians $\mathcal{G}^S = \{(\boldsymbol{\mu}_i, \boldsymbol{\Sigma}_i, \alpha_i, \mathbf{c}_i)\}_{i=1}^{N_S}$, where each Gaussian has center $\boldsymbol{\mu}_i \in \mathbb{R}^3$, covariance $\boldsymbol{\Sigma}_i \in \mathbb{R}^{3\times3}$, opacity $\alpha_i \in (0,1)$, and possibly view-dependent color $\mathbf{c}_i$.

Under the camera projection $\Pi_t$, each Gaussian projects to an ellipse with screen-space covariance $\boldsymbol{\Sigma}_{i,t}^{2D} = J_t \boldsymbol{\Sigma}_i J_t^\top$, where $J_t = \frac{\partial \Pi_t}{\partial \mathbf{x}}\big|_{\boldsymbol{\mu}_i}$ and $\mathbf{u}_i = \Pi_t(\boldsymbol{\mu}_i)$. Its pixel contribution at $\mathbf{u}$ is $g_{i,t}(\mathbf{u}) = \exp\left(-\frac{1}{2}(\mathbf{u}-\mathbf{u}_i)^\top (\boldsymbol{\Sigma}_{i,t}^{2D})^{-1}(\mathbf{u}-\mathbf{u}_i)\right)$, yielding effective opacity $\hat{\alpha}_{i,t}(\mathbf{u}) = \alpha_i \, g_{i,t}(\mathbf{u})$. The rendered image is obtained via front-to-back alpha compositing,

$$\hat{\mathbf{I}}_t(\mathbf{u}) = \sum_{i \in \mathcal{S}_t(\mathbf{u})} \Big( \prod_{j \in \mathcal{S}_t(\mathbf{u}),\, j < i} \big(1 - \hat{\alpha}_{j,t}(\mathbf{u})\big) \Big) \hat{\alpha}_{i,t}(\mathbf{u}) \, \mathbf{c}_i,$$

where $\mathcal{S}_t(\mathbf{u})$ denotes the depth-sorted splats overlapping pixel $\mathbf{u}$. Parameters $\Theta = \{\boldsymbol{\mu}_i, \boldsymbol{\Sigma}_i, \alpha_i, \mathbf{c}_i\}$ are optimized with the standard 3DGS photometric loss across frames.

**Human Gaussians.** We learn deformable human Gaussian representations from multi-view images that can be animated with different SMPL poses. Our approach consists of three key steps:

*Step 1: Canonical Gaussian parameterization on SMPL.* Given multi-view images of a person performing diverse poses, we learn a mapping from SMPL poses to 3D Gaussians in posed space. Following (Li et al., 2023b), we place Gaussians on the 2D manifold of the SMPL surface by constructing an approximate front–back UV atlas via orthographic projections of the SMPL mesh. Let $\boldsymbol{\beta}$ denote SMPL shape and $\boldsymbol{\theta}_t$ the pose at time $t$. We rasterize pose-conditioned features into a *pose map* $P_t \in \mathbb{R}^{H \times W \times C}$ - denoted using $\mathcal{M}(\boldsymbol{\beta}, \boldsymbol{\theta_t})$. A per-identity StyleUNet $f_\phi$ predicts a set of *canonical human Gaussians* anchored on the SMPL surface: $\mathcal{G}_t^{\mathsf{C}} = f_\phi(P_t) = \big\{(\mathbf{x}_k^{\mathsf{C}}, \boldsymbol{\Sigma}_k^{\mathsf{C}}, \mathbf{c}_k, \alpha_k)\big\}_{k=1}^{N_H}$.

*Step 2: Skinning to posed space (LBS).* We obtain *posed* Gaussians by applying linear blend skinning (LBS) to canonical Gaussians $\mathcal{G}_t^C$ with SMPL joint transformations $\{(\mathbf{R}_b(\boldsymbol{\theta}_t), \mathbf{t}_b(\boldsymbol{\theta}_t))\}_{b=1}^B$ and vertex/bone weights $w_{kb}$ inherited from the SMPL surface by using nearest neighbour sampling from Canonical Gaussians to SMPL vertices. With $\mathcal{G}_t^{\mathsf{P}} = \{(\mathbf{x}_k^{\mathsf{P}}, \boldsymbol{\Sigma}_k^{\mathsf{P}}, \alpha_k, \mathbf{c}_k)\}_{k=1}^{N_H}$ we denote the posed human Gaussians at time $t$. During training, we render these posed Gaussians using standard 3DGS and supervise using multi-view images and cameras.

*Step 3: Pose-to-Gaussian inference (test-time).* Following (Li et al., 2023b), we compute the top $K \in [10, 20]$ PCA components of training pose maps $\{P_t\}$, yielding mean $\bar{P}$ and basis $Q_K$. At inference, poses synthesized by our Gaussian-aligned motion module (Sec. 3.2) are first projected to this subspace and then mapped to posed Gaussians: $\tilde{P}_y = \bar{P} + Q_K z_y \quad \mathcal{G}_y^{\mathsf{P}} = \mathrm{LBS}_{\boldsymbol{\theta}_y}\big(f_\phi(\pi(\mathcal{M}(\beta, \boldsymbol{\theta}_y)))\big)$. Here we use $\pi$ to denote the projection to the subspace operation and $y$ to indicate a test-time pose. For further details please see supplementary materials.

## 3.2 GAUSSIAN-ALIGNED MOTION SYNTHESIS

We introduce a Gaussian-aligned motion module that synthesizes controllable human motion directly in 3DGS scenes. Our key novelty is twofold: (i) we deploy reinforcement learning (RL) in Gaussian space by deriving reliable scene cues from opacity-weighted projections (no meshes or paired human–scene data required); and (ii) we couple RL locomotion with a deterministic latent optimizer for precise, contact-sensitive transitions in 3DGS scenes.

**Design overview.** We reuse a strong latent motion backbone trained on large scale mocap dataset and add two 3DGS-specific controllers: an RL locomotion policy that navigates between waypoints while avoiding scene Gaussians, and a deterministic latent-space optimizer that executes short, fine-grained actions near targets (e.g., stop, sit, grasp) before returning control to RL. While this explicit decomposition is not typical in existing motion synthesis frameworks, we find it especially effective in 3DGS settings, as this allows us to exploit the fact that in 3DGS scenes much of the raw scene detail can be abstracted to (i) a set of *paths* for navigation and (ii) *action points* (e.g., sitting locations, grasping targets provided by an animator) at which specific behaviors are executed - thus allowing for scene-aware motion synthesis without human-scene paired data. Both submodules operate consistently in the latent space of a learned motion model (Zhao et al., 2025).

**Latent motion backbone.** We adopt a latent motion prior, following prior work (Zhao et al., 2025) trained on AMASS (Punnakkal et al., 2021; Mahmood et al., 2019). Specifically, the model learns a compact motion-primitive space with a transformer VAE trained on mocap data, and places a diffusion prior in this latent space. Given motion history $\mathbf{H}$ and a future motion segment $\mathbf{X}$, the encoder $\mathcal{E}$ outputs a Gaussian posterior $q_\phi(\mathbf{z} \mid \mathbf{H}, \mathbf{X}) = \mathcal{N}(\boldsymbol{\mu}, \sigma^2 \mathbf{I})$ with reparameterized sample $\mathbf{z} = \boldsymbol{\mu} + \sigma \odot \boldsymbol{\varepsilon}$ where $\boldsymbol{\varepsilon} \sim \mathcal{N}(\mathbf{0}, \mathbf{I})$. The decoder $\mathcal{D}$ reconstructs motion as $\hat{\mathbf{X}} = \mathcal{D}(\mathbf{H}, \mathbf{z})$. On this

latent space, a denoiser $\mathcal{G}$ operates with forward process

$$q(\mathbf{z}_\tau \mid \mathbf{z}_{\tau-1}) = \mathcal{N}(\sqrt{1 - \beta_\tau}\, \mathbf{z}_{\tau-1},\, \beta_\tau \mathbf{I})$$

and predicts the clean code $\widehat{\mathbf{z}}_0 = \mathcal{G}(\mathbf{z}_\tau, \tau, \mathbf{H}, \mathbf{c})$, where $\mathbf{c}$ is an optional text embedding. During inference we sample $\mathbf{z}_{\tau_{\max}} \sim \mathcal{N}(\mathbf{0}, \mathbf{I})$, perform about $\tau_{\max} \approx 10$ denoising steps to obtain $\widehat{\mathbf{z}}_0$, decode $\widehat{\mathbf{X}} = \mathcal{D}(\mathbf{H}, \widehat{\mathbf{z}}_0)$, and update $\mathbf{H}$ with the last $H$ frames for autoregressive sampling. This latent backbone is reused; our contribution lies in coupling it with 3DGS-specific controllers.

**Scene-adapted RL locomotion in 3DGS.** Our insight here is that locomotion policies trained in mesh-based synthetic environments (Zhao et al., 2023) can be used in 3DGS reconstructions when combined with our scene adaptation. We cast navigation as an MDP whose *action space* is the latent space of the motion model. The policy outputs a start-noise $\mathbf{z}_{\mathrm{RL},i}^{(\tau_{\max})}$, which a frozen $\mathcal{G}, \mathcal{D}$ map to a short motion clip, ensuring stable rollouts. At step $i$, the agent observes state $s_i = (\mathbf{H}_i, \mathbf{g}_i, \boldsymbol{\eta}_i, \mathbf{c}_i)$ where $\mathbf{H}_i$ is motion history, $\mathbf{g}_i$ a goal cue, $\boldsymbol{\eta}_i$ a scene cue, and $\mathbf{c}_i$ a text embedding. The policy samples $a_i \sim \pi_\theta(\cdot \mid s_i)$, interpreted as $\mathbf{z}_{\mathrm{RL},i}^{(\tau_{\max})}$. The resulting clip $\mathbf{X}_i$ updates the history $\mathbf{H}_{i+1}$. Rewards $r_i = r(s_i, a_i, s_{i+1})$ encourage waypoint progress, obstacle avoidance, and kinematic plausibility. Training follows synthetic mesh-based environments as in (Zhao et al., 2023), while our contribution is the deployment in 3DGS. For deployment in 3DGS scenes (which lack meshes), we approximate navigation regions via orthographic projection: (i) compute PCA of Gaussian centers to align a top-down view, (ii) threshold opacities to filter floaters, (iii) render a binary map of obstacles and run A* for pathfinding. The policy consumes an egocentric occupancy grid/walkability map $\mathcal{M} \in \{0,1\}^{N \times N}$ centered on the agent. For each grid cell $u$, we compute its nearest-neighbor distance $d(u)$ to filtered Gaussians and mark $\mathcal{M}(u) = 1$ if $d(u) > \tau$, else 0. Despite being approximate, this provides sufficiently reliable local context for navigation in 3DGS scenes. For inference during locomotion, we fix text cue $\mathbf{c}_i$ to "walk". For details please see supp mat.

**Latent optimization for transitions in 3DGS.** Once the agent reaches the vicinity of an action point, control switches from RL to *deterministic latent-space optimization* for fine-grained actions such as stopping, sitting, or grasping. Following (Zhao et al., 2025), we adopt a deterministic DDIM sampler (no step-skipping), which defines a fixed rollout (see supp. mat.) $\mathbf{M} = \mathrm{ROLLOUT}(\mathbf{Z}_{\mathrm{opt}}, \mathbf{H}_{\mathrm{seed}}, \mathbf{C})$, where $\mathbf{Z}_{\mathrm{opt}}$ is the terminal noise variable, $\mathbf{H}_{\mathrm{seed}}$ the seed history, and $\mathbf{C}$ a fixed text cue ("sit", "grab"). We optimize $\mathbf{Z}_{\mathrm{opt}}$ by minimizing

$$\mathcal{L}(\mathbf{Z}_{\mathrm{opt}}) = F(\Pi(\mathbf{M}), \mathbf{g}_{\mathbf{user}}) + \mathrm{Cons}(\mathbf{M})$$

with gradient updates $\mathbf{Z}_{\mathrm{opt}}^{(k+1)} = \mathbf{Z}_{\mathrm{opt}}^{(k)} - \eta \nabla_{\mathbf{Z}_{\mathrm{opt}}} \mathcal{L}(\mathbf{Z}_{\mathrm{opt}}^{(k)})$, where $\Pi(\cdot)$ projects the rollout onto task-relevant variables, $F$ measures goal satisfaction, and Cons adds continuity, collision, and smoothness constraints.

For position-only goals $\mathbf{g}_{\mathbf{user}}$ (e.g., sitting or grabbing at a user-provided point), we synthesize a short $f$-frame snippet $\mathbf{M} = (\mathbf{M}_1, \ldots, \mathbf{M}_f)$ starting from the locomotion end state $\mathbf{M}_{\mathrm{end}}^{\mathrm{loc}}$. In this setting, $F$ corresponds to the reach and stop terms, with $\mathcal{L}_{\mathrm{reach}} = \|\mathbf{x}_f(j^\star) - \mathbf{g}\|_2^2$ and $\mathcal{L}_{\mathrm{stop}} = \|\mathbf{v}_f(j^\star)\|_2^2$, while Cons corresponds to start-continuity $\mathcal{L}_{\mathrm{start}} = \|\mathbf{M}_1 - \mathbf{M}_{\mathrm{end}}^{\mathrm{loc}}\|_2^2$, collision $\mathcal{L}_{\mathrm{coll}} = \sum_{b \in \mathcal{B}_f}[-\phi(b)]_+^2$, and smoothness $\mathcal{L}_{\mathrm{smooth}} = \frac{1}{f-1}\sum_{t=2}^{f} \|\mathbf{M}_t - \mathbf{M}_{t-1}\|_2^2$. The final objective is therefore

$$\mathcal{L} = \mathcal{L}_{\mathrm{reach}} + \lambda_v \mathcal{L}_{\mathrm{stop}} + \lambda_{\mathrm{start}} \mathcal{L}_{\mathrm{start}} + \lambda_{\mathrm{coll}} \mathcal{L}_{\mathrm{coll}} + \lambda_s \mathcal{L}_{\mathrm{smooth}}$$

Here $j^\star$ is an anchor joint (e.g., pelvis), $\mathbf{x}_f, \mathbf{v}_f$ are its pose and velocity at frame $f$, $\mathcal{B}_f$ are sampled SMPL points at frame $f$, and $\phi(\cdot)$ is a differentiable signed-distance proxy to 3DGS Gaussians . After completing the action, the same formulation synthesizes the exit transition (e.g., sit $\rightarrow$ walk), after which locomotion resumes. Though our method can synthesize actions which mimic "picking up", or "grabbing" in the scene it cannot lift actual objects from the scene - as we assume that the scene remains static throughout.

## 3.3 DIFFERENTIABLE CONTACT REFINEMENT IN 3DGS

After animating the reconstructed human Gaussians with synthesized human motion data (Sec. 3.2), we place them into the reconstructed 3DGS scene. Here we want to highlight that we never use the SMPL mesh. We use the "SMPL pose" (the joint angles of the 24 SMPL joints or the 54

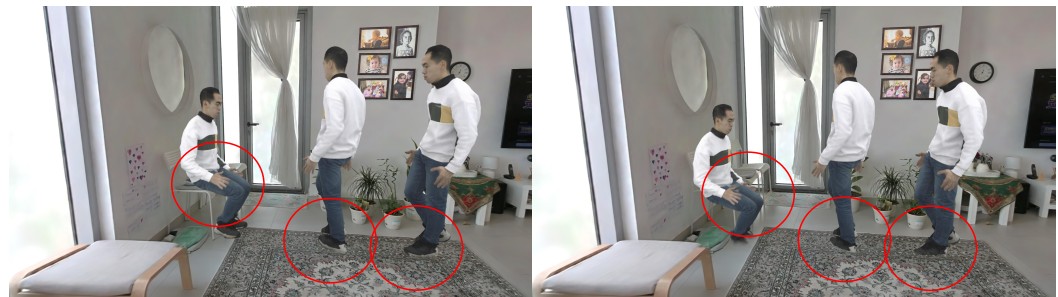

Figure 3: With (left) and right (without) refinment of Gaussians

SMPL-X joints) - to drive the 3DGS avatar. A naive composition of posed human Gaussians with scene Gaussians often leads to floor/geometry penetration and inconsistent contacts. We introduce a contact-aware refinement that solves for small, physically meaningful translations of a sparse set of human Gaussians so that contacts are respected and penetrations are reduced. (Fig. 3)

We first note that mapping SMPL pose to 3D Gaussians is fundamentally different from mapping SMPL pose to 3D SMPL mesh. Mapping SMPL pose to a mesh is just a linear blend skinning operation that maps canonical SMPL mesh vertices to posed SMPL vertices - hence the SMPL pose can usually be easily optimized to maintain contact with the scene. This is not the case with 3D human Gaussians. Mapping SMPL pose to 3D human Gaussians involves a forward pass through the learnt network - and then applying LBS to the output Gaussians. If we were to optimize the SMPL pose for correct contact this would entail getting a gradient through the neural network that maps SMPL pose to 3D Gaussians - which would probably be difficult to converge. Instead we optimize per-Gaussian offsets as the Gaussians (after being output by the network) are already placed close enough to reasonable locations in the 3d scene; thus we can simply optimize per gaussian offsets - with heavy regularization for correct contact. We describe the setup in detail below.

**Setup.** Let the posed human Gaussians at time $t$ be $\mathcal{G}_t^{\mathsf{P}} = \{(\mathbf{x}_k^{\mathsf{P}}, \mathbf{\Sigma}_k^{\mathsf{P}}, \alpha_k, \mathbf{c}_k)\}_{k=1}^{N_H}$, and the scene Gaussians be $\mathcal{G}^S$. Our goal is to refine a subset of the human Gaussians by per-frame translations $\mathbf{T}_{k,t}$ to achieve (i) contact where appropriate and (ii) separation elsewhere.

**Contact detection and indexing.** From synthesized SMPL motion, we detect contact frames for a set of body joints using simple kinematic cues. For joint $c$ with position $\mathbf{p}_{c,t}$, velocity $\mathbf{v}_{c,t} = \mathbf{p}_{c,t} - \mathbf{p}_{c,t-1}$ and acceleration $\mathbf{a}_{c,t} = \mathbf{v}_{c,t} - \mathbf{v}_{c,t-1}$, a frame is marked as contact if $\delta_{c,t} = (|v_{c,t}^y| < \tau_v) \wedge (a_{c,t}^y < \tau_a)$, where $y$ is the vertical axis. Because human Gaussian templates have identity-dependent counts and no global correspondence, we lift SMPL contact vertices $V_c^{\mathrm{SMPL}}$ (e.g., feet, hip) to the human Gaussians via nearest-neighbour search in the canonical space: $i^\star = \arg\min_k \|\mathbf{x}_k^{\mathsf{C}} - \mathbf{u}\|_2, \mathbf{u} \in V_c^{\mathrm{SMPL}}$. The resulting index set $\mathcal{I}_c$ specifies which human Gaussians may be refined at contact.

**Scene proximity in Gaussian space.** We measure scene proximity using a soft nearest-neighbour distance to scene Gaussians

$$d_\beta(\mathbf{x}) = -\tfrac{1}{\beta} \log\Big( \sum_{j=1}^{N_S} \exp\big( -\beta \|\mathbf{x} - \boldsymbol{\mu}_j\|\big)\Big),$$

where $\boldsymbol{\mu}_j$ are scene Gaussian centers and $\beta$ controls softness. This provides stable gradients for contact/separation without requiring explicit meshes.

**Refinement objective.** For a contact Gaussian $k \in \mathcal{I}_c$ at frame $t$ with indicator $\delta_{c,t}$, we optimize a translation $\mathbf{T}_{k,t}$ and update $\tilde{\mathbf{x}}_{k,t}^{\mathsf{P}} = \mathbf{x}_{k,t}^{\mathsf{P}} + \mathbf{T}_{k,t}$ by minimizing

$$\mathbf{T}_{k,t}^\star = \arg\min_{\mathbf{T}} \lambda_s \|\mathbf{x}_{k,t}^{\mathsf{P}} + \mathbf{T} - \boldsymbol{\mu}_{j(k,t)}\|_2^2 + \lambda_d \psi(\mathbf{x}_{k,t}^{\mathsf{P}} + \mathbf{T}, \delta_{c,t}) + \lambda_r \|\mathbf{T}\|_2^2,$$

where $\boldsymbol{\mu}_{j(k,t)}$ is the nearest scene Gaussian center and

$$\psi(\mathbf{x}, \delta) = \begin{cases} d_\beta(\mathbf{x})^2, & \delta = 1 \text{ (enforce contact)} \\ h_r\big(d_\beta(\mathbf{x})\big)^2, & \delta = 0 \text{ (enforce separation)} \end{cases} \quad \text{with } h_r(d) = \max(0, r - d).$$

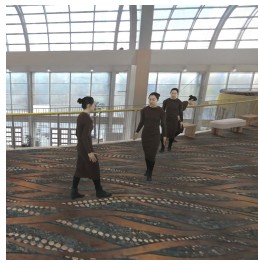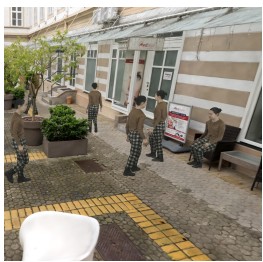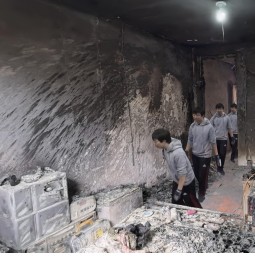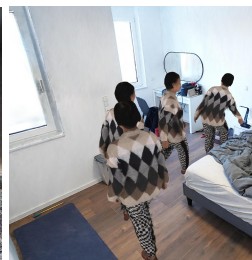

Figure 4: Qualitative results: Our method generates diverse motions across scenes and identities.

Table 1: **Evaluation design.** Two baselines × two protocols. HQ: highest-quality rendering settings for each method. The same camera trajectories are used within each pairwise comparison.

| Setting | Dataset / Source | 3DGS Scene (ours) | Recons Mesh Scene (Baseline) | Baseline Rendering | Protocols |
|---|---|---|---|---|---|
| **Baseline A** | Mon. Vids (same scenes) | 3DGS reconstruction | VGGT dense | 3DGS | I and II |
| **Baseline B** | Replica and Curated | 3DGS SuperSplat | Replica | Mesh | I and II |

For temporal coherence, we add $\lambda_t \sum_t \|\mathbf{T}_{k,t} - \mathbf{T}_{k,t-1}\|_2^2$. Intuitively, the objective snaps designated contact Gaussians toward nearby scene surfaces when contact is detected, pushes them away otherwise, penalizes large displacements, and smooths motion over time.

The refined human Gaussians $\tilde{\mathcal{G}}_t^{\mathsf{P}} = \{(\tilde{\mathbf{x}}_{k,t}^{\mathsf{P}}, \boldsymbol{\Sigma}_k^{\mathsf{P}}, \alpha_k, \mathbf{c}_k)\}_{k=1}^{N_H}$ are composed with $\mathcal{G}^S$ and rendered with the standard 3DGS rasterizer to produce photorealistic interactions (e.g., walking, sitting) with improved contact fidelity and fewer penetrations. To the best of our knowledge, this is the first mesh-free refinement in Gaussian space that leverages a differentiable scene-distance, remains identity-agnostic via SMPL-to-Gaussian lifting, and operates as a lightweight post-hoc stage to improve contact realism without retraining.

## 4 EXPERIMENTS

For rendering evaluation, we present two modified mesh-based baselines (Baseline A and B) and evaluate with two evaluation protocols (I-human and II-automated). For further evaluation on motion quality and ablations please see the supplementary.

### 4.1 RENDERING EVALUATION

**Baseline A:** For Baseline A we collect monocular videos from DL3DV (Ling et al., 2024); each scene is reconstructed twice (once as 3DGS, once as a mesh using dense VGGT reconstruction (Wang et al., 2025)) so that comparisons are *within-scene*. Using the meshes obtained using dense VGGT reconstruction, we again use (Zhao et al., 2023) to generate SMPL-X parameters. Then we use these parameters naively to pose human Gaussians (Sec. 3.1) in the 3D scene and render the composited scene and human Gaussians using 3DGS. Note we do not perform any refinement. Furthermore note that the scene mesh is only used for motion synthesis but for rendering we use the 3DGS scene reconstruction and the posed human Gaussians. Baseline A is designed to show that a naive baseline that composites human and scenes Gaussians does not work out-of-the-box for monocular videos and hence provides further motivation for our algorithm. For baseline A evaluations, we render synchronized camera trajectories per pair (identical poses, FoV, and exposure).

**Baseline B:** In this experiment we aim to evaluate the rendering quality of a strong mesh-based baseline vs our 3DGS based algorithm. We use the highest quality existing mesh based 3D scenes from the Replica (Straub et al., 2019) dataset. In the Replica Scene we use the framework in (Zhao et al., 2023) to synthesize motion. Then we use a rigged scan from RenderPeople (along with its texture map) (RenderPeople) animated with the synthesized motion parameters in the Replica scene to generate the final renderings. For rendering our videos we use scenes from the Supersplat library and Avatars from Avatarrex (Zheng et al., 2023) dataset. This experiment aims to evaluate the highest

Table 2: **Human preference study (win rate, %)** — fraction of pairwise trials where OURS is preferred. Baseline B compares OURS vs a mesh based baseline at highest-quality; Baseline A compares OURS (3DGS) vs a custom baseline designed for monocular videos. Higher is better.

| | Replica vs 3DGS-Library (Baseline B) | Monocular (Baseline A) |
|---|---|---|
| OURS(3DGS) VS MESH | 82.1 | 72.9 |

Table 3: **VLM preference study (win rate, %)** — fraction of pairwise comparisons where OURS is preferred. Baseline B compares OURS vs a mesh based baseline at highest-quality; Baseline A compares OURS (3DGS) vs a custom baseline designed for monocular videos. Higher is better.

| | Replica vs 3DGS-Library (Baseline B) | Monocular (Baseline A) |
|---|---|---|
| GPT-5 | 75.2 | 71.8 |
| Gemini 2.5 | 69.1 | 65.9 |

quality rendering of a mesh based rendering vs a highest quality 3DGS renderings for the specific setting of human-scene animation. Note comparisons are not within scene.

Here we want to highlight that finding the same scene+human combination for both our method and Baseline B would be difficult - because the data capture pipelines for 1) 3DGS vs mesh scene 2) 3DGS vs mesh human are fundamentally different. RenderPeople uses a rig of 250 DSLR cameras to capture a 3D human mesh - while the AvatarX dataset only uses 16 cameras and Behave dataset uses only 4. For 3DGS Avatar reconstruction motion of about 120 seconds inside the multiview camera setup is required while for a mesh capture only one frame is required. The way 3D scenes are reconstructed for Replica (mesh) and SuperSplat (3DGS) are also fundamentally - the replica reconstruction uses depth while some supersplat scenes use lidar. Additionally the replica original images are not available so we can't reconstruct a 3DGS splat for Replica scenes of the same quality as the ones on SuperSplat. As such for this particular baselines comparisons are not within scene.

We also want to acknowledge that for 3DGS animatable Avatars we require multiview video while a mesh can be reconstructed using multiview images. However we believe that for different use-cases, users would be willing to make the tradeoff for higher rendering quality.

**Evaluation Protocol I: Human preference study** We conduct a pairwise *forced-choice* study measuring perceived photorealism. Each trial presents two *mute* videos from ours vs Baseline A or ours vs Baseline B. Participants select the video they find *more photorealistic*. We generate 5 samples for both comparisons and aggregate votes by comparison. We collect 21 participants for both Baseline A and B. We report *win rate (%)* of OURS over its comparator.

**Evaluation Protocol II: VLM-based pairwise judgment** We use two strong vision–language model (VLM): GPT-5 (OpenAI, 2025) and Gemini2.5 (Comanici et al., 2025) as *paired comparators* between still renderings. For each video pair, we uniformly sample 10 frames per method, form matched pairs at the same timestamps, and query the VLM with "which image looks more photoreal?". The VLM outputs a ternary judgment {Left wins, Right wins, Tie}; we compute a *VLM win rate* as the percentage of non-tied pairs favoring OURS. We randomize image order and prevent leakage by removing textual overlays and metadata.

For the pairwise VLM study, the VLM is instructed to ignore artistic style and focus on physical plausibility: *"which image looks more photoreal? Consider geometry (straight lines, depth cues), materials (BRDF, speculars), lighting/shadows, and absence of artifacts (flicker, halos, floaters).*

**Results Baseline A (Same monocular video)** On the within-scene comparison (Fig. 6), OURS outperforms the mesh baseline in both human and VLM judgments (Tables 2–3). Note that the scene meshes reconstructed using VGGT often exhibit blocky strucutres, blocked paths and hence are not suitable for motion synthesis, while our algorithm directly operates in Gaussian Space and doesnt suffer from the same problems. For detailed failure modes see Supp Mat.

**Baseline B (Replica Scene), HQ vs HQ.** Across both evaluations (Fig. 6) OURS is preferred by humans and by the VLM comparator (Tables 2–3) - thus clearly underscoring the central premise

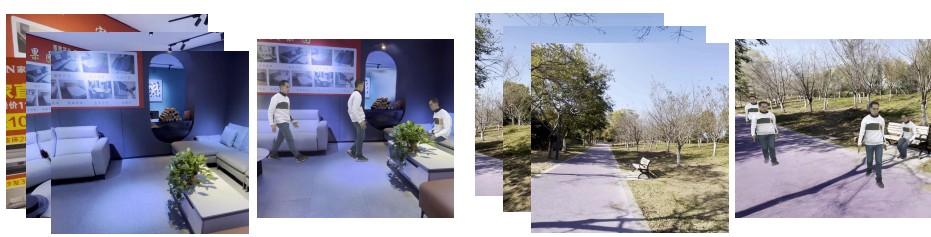

Figure 5: Free viewpoint rendering of edited monocular video with animated humans

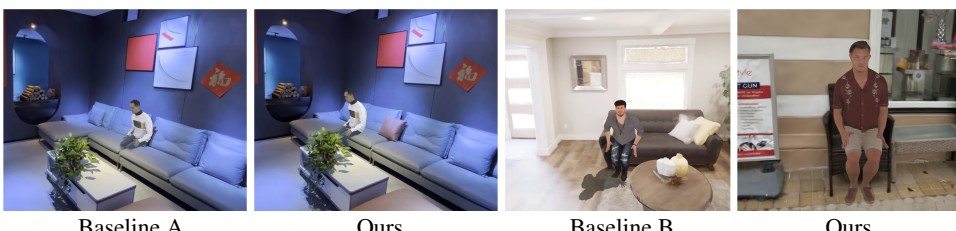

| Baseline A | Ours | Baseline B | Ours |

Figure 6: Visual Comparisons with Baselines

of the paper - *that neural scene representations yield better rendering quality for human-scene interaction* compared to existing mesh based representations.

## 4.2 QUALITATIVE RESULTS AND FREE VIEWPOINT RENDERING OF EDITED VIDEOS

In Fig. 4, we show results for diverse scenes from the SuperSplat library, Scannet scenes with Avatars from BEHAVE (sparse only 4 cameras), DNA-Rendering (Cheng et al., 2023), Avatarrex (Zheng et al., 2023) datasets. In Fig. 5, we demonstrate that our method works for monocular RGB scene videos and allows for free viewpoint rendering of videos edited with geometry consistent placement of animated humans in the scene. For more results please see the supplementary.

## 5 CONCLUSION

We have presented, to the best of our knowledge, the first method to synthesize human interactions in diverse 3D environments using 3D Gaussian Splatting (3DGS) as the underlying 3D representation. Our results suggest that neural rendering is now mature enough to function as a practical component in end-to-end 3D human–scene animation pipelines, bridging previously disjoint lines of work in human-scene animation and neural rendering. **Crucially, our pipeline operates on scenes reconstructed from monocular RGB video** and allows for applications such as monocular RGB geometry consistent video editing. We believe this framing and evidence open new research directions at the intersection of human animation, scene understanding, and neural rendering.

**Limitations.** Despite this progress, our pipeline has several limitations. First, complex and rapidly changing illumination can cause rendering artifacts and imperfect relighting. Second, we do not enforce full physics-based constraints, which can yield interactions that look plausible yet violate contact, stability, or momentum conservation. Third, the range of interaction types is limited; highly dexterous manipulation and long-horizon, multi-contact behaviors remain challenging. Fourth, we assume access to multiview videos of a human performing diverse actions.

**Outlook.** Addressing these issues suggests several promising research directions: integrating stronger lighting estimation and inverse rendering, incorporating differentiable or learned physics priors for contact and dynamics, expanding the interaction vocabulary to richer, longer, and multi-person scenarios and investigation how Avatars that generalize to Out-of-distribution poses can be reconstructed from monocular videos. We hope this work provides a foundation for scalable, video-native human–scene animation pipelines and catalyzes further advances in data, models, and evaluation for interactive 3D human animation.

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
