# OpenReview forum: "AHA! Animating Human Avatars in Diverse Scenes with Gaussian Splatting"
_ICLR.cc/2026/Conference — Submitted to ICLR 2026_

### Official Review · Reviewer_2xxi · 2025-10-21

**Soundness:** 3
**Presentation:** 3
**Contribution:** 3
**Rating:** 6
**Confidence:** 4

**Summary:**

This paper presents the successful integration of a reconstructed, drivable 3D Gaussian human avatar into a 3D Gaussian Splatting (3DGS) scene, enabling fundamental actions like walking and sitting.
The core contribution is the effective migration and adaptation of traditional scene-mesh human interaction techniques (including RL-based locomotion and motion transitions) to the 3DGS. This required specific technical adjustments, notably: Gaussian point simplification (via PCA) and floating point filtering for rendering a cleaner Gaussian obstacle map needed for path planning. Furthermore, the work includes contact optimization to address foot-ground interaction issues.
Overall, this work is a meaningful contribution to the community, expanding the potential application scenarios for 3DGS avatars.

**Strengths:**

1.Achieved a novel and effective integration of a human avatar into a realistic 3DGS environment, successfully demonstrating basic locomotion (walking) and sitting actions.
2.The paper tackles key challenges inherent in porting RL locomotion from structured mesh scenes to  3DGS. The use of PCA for simplification and floating point filtering effectively reduces noise in the obstacle perception, which is crucial for path planning.
3.The inclusion of contact optimization is valuable, as it helps mitigate common artifacts like foot sliding, floating, or unnatural foot interaction with the ground plane, thus enhancing overall realism.

**Weaknesses:**

1.The experimental section lacks comprehensive visual validation. The paper only showcases the results of the proposed method without providing visual comparisons corresponding to Baseline A and Baseline B.
2.The work appears to overlook a crucial aspect of realism when integrating an avatar: lighting and shadows. An important requirement for placing a human into a varying scene environment is that the avatar's appearance (including self-shadows and scene-induced illumination changes) should react naturally to the ambient light, which is currently a strength of mesh-based approaches. This omission significantly limits the potential for higher evaluation.
3.Minor foot clipping briefly observable in the accompanying video around the 1:20 mark, suggesting the contact optimization could still benefit from further refinement.
4.The method's pipeline diagram does not clearly highlight the core technical contribution or the novel, adapted aspects of the proposed approach.

**Questions:**

1.Could the authors clarify why the video only shows the comparison between Baseline A vs. Ours, and not include the visual comparison for Baseline B vs. Ours?
2.It would be highly beneficial to include a visual comparison showing the achieved result side-by-side with a simple real-world reference video (e.g., the same person performing the same walking or sitting actions in the same scene) to better gauge the fidelity and realism.
3.Please elaborate on the specific method or process used to accurately handle the scale and proportional alignment of the avatar when placing it into the reconstructed 3DGS scene.
4.After the initial scene and avatar reconstruction, what is the approximate training or optimization time required for the proposed method to successfully enable path-planned walking and sitting within the scene?
5.Beyond simple walking, can the proposed method support more dynamic and large-scale motions, such as running, jumping, or dancing?
6.The paper mentions the "picking up" action. Based on the description and lack of scene object segmentation, achieving this action seems highly challenging, if not currently infeasible. We suggest removing or clarifying this claim to prevent potential misunderstanding regarding the system's capabilities

---

> ### Author Response · Authors · 2025-11-17
>
> We thank the reviewer for their feedback. We are highly encouraged to see that they recognize the novelty of our idea and the quality of our results.
>
> **1) Visual comparison**: We invite the reviewer to watch the supplementary video at timestep 3:43 to 4:48 for visual comparison. We have also updated the manuscript to include visual comparisons which were previously only included in supp video. We thank the reviewer for this feedback.
>
> **2) Shadows and lighting**: We acknowledge that shadows and lighting are not addressed - we write so ourselves in the conclusion. However we want to highlight that this is the first method that tries to address human animation in 3d scenes using neural rendering instead of classical rendering. We believe this should provide others in the community a strong baseline to build upon. Also despite this limitation our renderings are still considered to be more photoreal, compared to a mesh-based rendering baseline, by a vast majority of the users in our user study (table 2)
>
> **3) Method figure:** We have updated the method figure. We hope it better reflects our contributions. We have also included a new figure in the paper which was previously in the supplementary to showcase our contact contribution better (fig 3). We thank the reviewer for this feedback. We hope the reviewer finds this new figure to be adequate. If the reviewer has further suggestions about the figure we would be happy to act upon those suggestions
>
> **4) Why no side-by-side for Baseline B:** As we write in the manuscript, it is impossible to obtain 3DGS scenes for scenes from the  Replica dataset (high quality scene dataset) or to obtain high quality meshes for scenes on the SuperSplat library (without losing quality).  As such comparisons with baseline B (mesh based rendering of high quality scenes) is not within scene (unlike BaselineA which is evaluated for same scenes)- so when asking users to evaluate this Baseline we simply ask them to evaluate general photorealism of the video. Hence when providing visualization for BaselineB we don’t provide side-by-side comparison as our results are already provided in the first part of the video - which serve as the benchmark for baselineB comparisons. This comparison serves to benchmark the rendering quality of 3DGS animation compared to the rendering quality of mesh-based animation. And as the user study indicates our method is considered to more photoreal by a vast majority of respondents.
>
> **5) Same person performing action in captured video** We appreciate this suggestion, however our method assumes that we have access to multiview RGB images of the Avatar; to execute this idea we would need a multiview scanner in which we could capture the person then capture their motion with a mobile video in a scene outside the scanner. Unfortunately we do not have a multiview scanner available and we only use publicly available multiview datasets - AvatarX, BEHAVE; but if the reviewer wishes we would be happy to include this idea in the manuscript about possible evaluations - in the conclusion. We appreciate the feedback.
>
> **6) Scale and proportion** We normalize all reconstructed Avatars to a fixed scale - i.e approximately 1.65 - by ensuring that the distance between max z and min z of the reconstructed Avatar is 1.65m . To ensure scenes have the same scale - we use the fact that scenes downloaded from SuperSplat have the same scale; Scannet++ scenes have the same scale and DL3DV have the same scale. We manually find the ratio that aligns the three datasets to one global scale where a 1.65m tall human would naturally fit.
>
> **7) Training or optimization time for path-planned** Over 5 motion sequences (on average 30 seconds) RL synthesis takes 10 seconds; optimization based synthesis takes 45 seconds (in total for transition in and out); contact optimization takes 60 seconds. In total for a 30 second sequence our whole algorithm takes approximately 300 seconds.
>
> **8) More motion:** As we write in the conclusion, the scope of motions that our method is capable of is limited - it can indeed walk, sit, get up and perform some grabbing action. We have tried to explicitly mention this in the conclusion. We hope our paper will serve as a catalyst for further research about diverse motion in 3d scenes with neural scene representations.
>
> **9) Picking up**: At timestep 1:58 we provide an example of a motion where the Avatar can mimic picking up something from the scene. i.e we can synthesize the action of “picking up” but not actually pick some dynamic object in the scene. If the reviewer would like us to clarify this further in the manuscript we would be happy to.
>
> We hope we have addressed all the concerns the reviewer raised in their original review;
> If there is anything the reviewer would like to see from us, for them to increase their rating further, we would be happy to provide further experiments, clarifications or update the manuscript as the reviewer desires.

---

> > ### Comment · Reviewer_2xxi · 2025-11-24
> > **Response to Authors**
> >
> > I thank the authors for their detailed response and for the efforts made to update the manuscript and supplementary video during the rebuttal period.
> >
> > "Picking Up" Action: Thank you for the clarification regarding the "picking up" motion. Please ensure that the distinction made in the rebuttal (that the avatar mimics the action without physically interacting with dynamic objects) is explicitly stated in the final camera-ready version to avoid misleading readers.
> >
> > Reproducibility: Finally, could the authors confirm if they plan to release the source code and data processing scripts upon acceptance? Given the complexity of the system (integrating RL, 3DGS, and contact optimization), open-sourcing the code would significantly enhance the contribution to the community and facilitate future research on this baseline.
> >
> > I will maintain my positive assessment of the paper.

---

> > > ### Author Response · Authors · 2025-11-26
> > >
> > > We thank the reviewer for their feedback. We have uploaded a new revision of the manuscript which incorporates the suggestions made by the reviewer. On L317 we explicitly mention for clarification that while our method can synthesize motions which mimic actions such as grabbing it cannot lift dynamic objects in the scene.  Regarding the code release, our institution mandates an internal legal review before any public open-sourcing. This is routine, and because our project is entirely research-oriented and contains no proprietary dependencies, we do not expect any issues. We are committed to releasing as much of the code we can following this standard approval process. If there is anything further - experiments or clarification or revisions - we can provide for the reviewer to increase their rating further to “accept” from "borderline accept”, we would be happy to do so.

---

### Official Review · Reviewer_CKuQ · 2025-10-24

**Soundness:** 1
**Presentation:** 2
**Contribution:** 2
**Rating:** 4
**Confidence:** 3

**Summary:**

The paper addresses the task of generating human motion within 3D environments while enabling photorealistic rendering of the resulting animation. To this end, the work proposes to utilize 3D Gaussian Splatting (3DGS) in contrast with meshes that are traditional for the motion generation task. The method decouples animation generation from 3DGS rendering and introduces a "refinement stage" to correct the Gaussians' placement in accordance with the animation. Experiments show visual improvements compared to two baselines: Baseline A - no "refinement stage"; Baseline B - mesh-based rendering.

**Strengths:**

* To the best of my knowledge, the work for the first time proposed to simultaneously solve the task of motion generation in the environment and photorealistic rendering.
* The work raises an interesting question: is it possible to generate human animation in the scene when both are presented with 3D Gaussians in contrast with meshes in previous works.
* The approach demonstrates convincing qualitative results with sufficient visual realism.

**Weaknesses:**

* Limited novelty. While the results are good, the presented method is mostly an engineering pipeline based on the existing methods with "refinement stage" as the only novel part. The refinement stage is presented as a set of heuristics to polish the Gaussians' positions: the frames and Gaussians are selected based on predefined rules and thresholds, as well as the Gaussians' offset directions.
* Continued reliance on meshes. While the work investigates the applicability of 3D Gaussians for motion in the scene generation, it still relies heavily on meshes. The animation of a person is generated utilizing SMPL parametric meshes in the "approximated navigation meshes" (line 296) as the environment. Thus, the work doesn't demonstrate the principal benefits of using Gaussians compared to meshes.
* Unclear method description. From the text, it is hard to tell what parts of the method are novel, because the work describes the utilized methods in too much detail. Describing only the necessary information on the existing methods in the "Preliminaries" subsection would help better separate the method from prior works.
* The experiments are not clearly presented. Table 1 is difficult to interpret, and the baselines are complex pipelines rather than previous works.

**Questions:**

* I would like to have more clarification on the choice of predicting offsets for each Gaussian instead of learning SMPL pose correction. Intuitively, it seems that adjusting the SMPL pose would result in fewer artifacts than the independent transformation of Gaussians.

---

> ### Author Response · Authors · 2025-11-17
>
> **1) What is novel? Why not just an “engineering pipeline”?**: We make 3 novel contributions 1) we show a neural scene representation 3DGS - as a 3D representation - can be used for human animation in 3d scenes 2) inside our Gaussian aligned motion module we show how to adapt RL-controllers which have been used for mesh-based scenes for scenes represented using 3DGS. As *reviewer 2xxi* writes we present “migration and adaptation of traditional scene-mesh human interaction techniques (including RL-based locomotion and motion transitions) to the 3DGS This required specific technical adjustments, notably: Gaussian point simplification (via PCA) and floating point filtering for rendering a cleaner Gaussian obstacle map needed for path planning.” We also show how to use 3DGS representation to compute a walkability map [l293] for navigation which is used with the RL based navigation - all these are technical ideas which go beyond existing mesh based navigation and motion synthesis 3) Furthermore while contact based optimization for human-scene interaction has indeed been used for mesh based representations, it has never ever been used in the context of neural rendering and we present the first effective use of such contact constraints for the 3DGS representation. Here we want to highlight that a Gaussian Avatar is fundamentally different from the SMPL mesh - in that every Gaussian Avatar has a different number of Gaussians - where as the number of vertices in the SMPL mesh always remains the same - we show how to effectively define Gaussians for hips, feet by transferring correspondences from SMPL to the Gaussian Avatar - again this technically goes beyond what has been shown for SMPL-scene contact optimization. We have updated the manuscript to include figures which were previously included in the supplementary material. The new figure 3 in the paper shows how contact optimization is necessary in 3dgs as without it the human penetrates through the chair for example.
>
> We also want to highlight that in the supplementary material - (Figure 1 and Figure 2 of supp mat) - we show what happens without our adaptations to path-finding and RL controllers  for navigation. We want to highlight that all prior work on motion synthesis works in 3D mesh space - several algorithms exist to compute obstacle free paths in meshes but we for the first time show what adaptations are necessary to apply these algorithms to a scene represented using 3DGS.We also want to highlight the ablation studies presented in the supplementary material. There we use scannet++ scenes to evaluate our motion synthesis adaptations - as scannet++ provides both mesh and 3dgs scenes. We add the table from the supp mat here -
>
>
> | Method                     | Foot Contact (cm/s) $\downarrow$ | Penetration (m) $\downarrow$ | Jerk (m/s$^3$) $\downarrow$ | Goal Reaching (\%) $\uparrow$ |
> |---------------------------|-----------------------------------|-------------------------------|------------------------------|--------------------------------|
> | **Ours (full)**           | 0.89                              | 0.012                         | 0.47                         | 95.2                           |
> | w/o opacity culling       | 1.45                              | 0.042                         | 0.78                         | 42.6                           |
> | w/o walkability map       | 1.37                              | 0.021                         | 0.71                         | 84.5                           |
> | w/o optimization refinement | 1.11                            | 0.016                         | 0.55                         | 91.3                           |
>
> As is clear without our opacity culling and walkability adaptations, motion when synthesized in 3dgs scenes is not natural.
>
> We also want to highlight Figure4 provided in the supplementary. Our method works with monocular RGB videos of scenes - this is impossible with any existing mesh based algorithm. To test how existing mesh based baselines work for RGB videos - we reconstruct a mesh from RGB video using VGGT. Figure 4 of supplementary showcases the typical failure modes of this reconstruction - there are artifacts blocking paths - and hence the baselines which rely on meshes fail. We render these results using an Avatar+3DGS. These are visualized in supp video from 3:40. There are clear failures hence showcasing the necessity of our adaptations for 3DGS scenes.

---

> ### Author Response · Authors · 2025-11-17
>
> **2) Reliance on meshes:** Please note that we never use the SMPL mesh. We use the “SMPL pose” (the joint angles of the 24 SMPL joints or the 54 SMPL-X joints) - never the “SMPL mesh” -  to drive the 3DGS avatar. We also never rely on either the scene mesh or the human mesh. When we say “approximate navigation meshes” - we use the word “navigation meshes” to indicate the 2D region of the world which is navigable - it is never ever an explicit mesh. To clarify any doubts we have updated the manuscript to indicate this. We thank the reviewer for this feedback.
>
> The 2D navigation area is obtained by projecting 3DGS using orthographic projection - with opacity based filtering. We again invite the reviewer to look at Figure 1 of the supp mat which provides a better idea of what the navigation region with 3DGS looks like.
> Again, we want to highlight throughout our results, we have never used the mesh representation for generating either the motion or the renderings or the path - For our method we have used 3DGS scenes reconstructed from monocular video or from the SuperSplat library and have presented the technical modifications necessary to make human animation work in 3DGS scenes.
>
> **3) Baselines and benefit of using gaussians over meshes**: As *reviewer gJYF* writes  “The authors attempted to design naive baselines (which I appreciate, since there were no previous 3D Gaussian works that enabled human-environment interaction), and the design is reasonable”.  Here we want to highlight that there are no existing methods for human animation in 3dgs scenes - hence we can't use any existing method straight out of the box. As such we have tried our level best to modify existing baselines to work for 3DGS scenes - or to make mesh based motion synthesis as photoreal as possible - this makes the baselines multi-layered systems instead of existing methods. If the reviewer would like us to compare our method with a particular specific baseline they have in mind we would be happy to provide the comparisons.
> In table 1 we have tried our best to summarize BaselineA, BaselineB and the evaluation protocols used. But we provide a much longer explanation of table 1 in the text in the beginning of the experiment section.
>
> For baseline B - we use an existing method for motion synthesis in 3D scenes represented as meshes - however this method just produces the SMPL mesh - which is not photoreal - while we focus on photorealism so we use the SMPL parameters produced by the method to animate a rigged 3D human scan from RenderPeople - represented as a mesh - hence the baseline becomes more than just using an existing method. Furthermore when we show naive raters our neural rendered results vs mesh based results - **82.1** prefer our results. Please see Table 2 in the paper. This also highlights that 3DGS is much more photoreal than a naive mesh based representation for animation - and is mature enough to act as a functional component of end-to-end animation pipelines.
>
> Another benefit of using 3dgs as a representation is that it allows for animation in scenes reconstructed using monocular videos - which is impossible with existing mesh based baselines.
>
> **4) Why optimize per-gaussian instead of SMPL pose for refinement:** We thank the author for this feedback. Please note that mapping SMPL pose -> 3D gaussians is fundamentally different from mapping SMPL pose -> 3D SMPL mesh. Mapping SMPL pose to a mesh is just a linear blend skinning operation that maps canonical SMPL mesh vertices to posed SMPL vertices - hence the SMPL pose can usually be easily optimized to maintain contact with the scene. This is not the case with 3D human gaussians. Mapping SMPL pose -> 3D human gaussians involves a forward pass through the learnt network  - and then applying LBS to the output Gaussians. If we were to optimize the SMPL pose for correct contact this would entail getting a gradient through the neural network that maps SMPL pose to 3D gaussians - which would probably be difficult to converge. Instead we optimize per-gaussian offsets - as the gaussians (after being output by the network) are already placed close enough to reasonable locations in the 3d scene - we can simply optimize per gaussian offsets - with heavy regularization - to make the human-scene gaussian contact correct. This also highlights the novelty of our contact constraint - 3D gaussians present a different challenge to a mesh based representation - and we have presented a contact constraint that works with a 3dgs representation. We hope this addresses the question.
>
> We thank the reviewer for their feedback. We hope we have addressed all the concerns raised by the reviewer. If there is anything else we can provide - clarification, new experiments, revision - for the reviewer to increase their rating - we would be happy to do so.

---

> > ### Comment · Reviewer_CKuQ · 2025-11-24
> >
> > The provided rebuttal satisfactorily addresses most of my concerns. Namely, the mesh usage, the choice of baselines, and the benefits of using 3DGS. But I still have concerns that the work is organized as an engineering pipeline that accumulates several off-the-shelf solutions with minor modifications to adapt them to 3DGS. Nevertheless, this work proposes to solve a novel task of photorealistic motion generation in the provided environment, and I believe it may be interesting to the community. Therefore, I'm leaning towards raising the rating to "Borderline accept".  I hope the authors will improve the final text to make the answers to the discussed questions more transparent in the paper.

---

> > > ### Author Response · Authors · 2025-11-26
> > >
> > > We thank the reviewer for upgrading their rating and we are grateful to them for their feedback. We have uploaded a new revision of the manuscript which addresses the questions raised by the reviewer. We have included the discussion about not using SMPL parameters and using per-gaussian optimization (L341). We have also included further clarification in the manuscript to explicitly mention that we do not use the SMPL mesh but only the SMPL pose for driving the 3DGS Avatar (L323). If there is anything further we can provide to address the concerns raised by the reviewer, we would be happy to do so.

---

### Official Review · Reviewer_b3rV · 2025-10-26

**Soundness:** 3
**Presentation:** 3
**Contribution:** 2
**Rating:** 4
**Confidence:** 3

**Summary:**

The paper tackles a new task that animates and renders a digital avatar in a 3D scene. While most of the previous works focus on one of animating digital avatars with existing poses, pose generation and 3D scene generation, this work combines them all as a unified pipeline.

The method starts with the reconstructions of digital avatars and 3D scenes seperately. Then it synthesizes the motions to plug the digital avatar into the scene. The motion synthesis incorporates a coarse module (an RL locomotion policy that predicts the waypoints) and a fine module (a deterministic latent-space optimizer that performs fine-grained motions when approaching the target). It follows with an optimization-based contact refinement to avoid penetration and enforce proper contact between the digital avatar and the scene.

It showcases the rendering results, particularly in the motions of walking and sitting. The motions are physically plausible with little penetration with the scenes.

**Strengths:**

* The task of animating digital avatars in 3D scenes is important in various AR/VR applications, e.g., game engines.
* Modeling 3D scenes and digital avatars seperately then combining them together in the test time is a reasonable choice which could be the use case in games where the digital avatars/3D scenes are replacable.
* The writing and the entire pipeline are easy to follow.

**Weaknesses:**

* The interactions demonstrated in the paper are limited to two simple motions—walking in open space and sitting on a chair. Other common daily activities, such as running or walking at different speeds, reaching for objects, or lying on a bed, are not explored.
* The motions shown in the supplementary videos, while physically plausible, still appear somewhat unnatural. For instance, the arms do not swing naturally or fluidly during walking sequences.
* Since the motion synthesis module is optimization-based, computational efficiency is a concern for real-world applications. For example, when controlling a digital avatar in a game, the system should be capable of planning and rendering motions in real time. It would be helpful if the paper reported the execution time of each module in the pipeline.
* The paper is largely a systematic study and lacks clear technical contributions. The individual components—such as 3D reconstruction of scenes and avatars—are well studied in prior work. Similarly, the RL-based locomotion policy and latent-space optimizer are established techniques in motion planning. The main technical novelty appears to lie in the contact refinement module, which extends ideas previously explored in SMPL-scene interactions to 3D Gaussian-scene interactions.

**Questions:**

* There is still cloudy artifact in the contact regions of feet and the ground. See 1:27-1:28 of the supplementary video, the contact of the feet and the mattress. In 1:36 of the supplementary video, when approaching the seats, the avatar performs a unnatural “leaning forward" motion. What is the cause of such artifacts in rendering and motion synthesis?

---

> ### Author Response · Authors · 2025-11-17
>
> We thank the reviewer for their comments.
>
> **1)What is novel? What is a technical contribution?**: We make 3 novel contributions 1) we show (for the first time) that a neural scene representation (3DGS) - as a 3D representation - can be used for human animation in 3d scenes 2) inside our Gaussian aligned motion module we show how to adapt RL-controllers which have been used for mesh-based scenes for scenes represented using 3DGS. As *reviewer 2xxi* - writes we present “migration and adaptation of traditional scene-mesh human interaction techniques (including RL-based locomotion and motion transitions) to the 3DGS This required specific technical adjustments, notably: Gaussian point simplification (via PCA) and floating point filtering for rendering a cleaner Gaussian obstacle map needed for path planning.” We also show how to use the 3DGS representation to compute a walkability map [l293] for navigation which is used with the RL based navigation - all these are technical ideas which go beyond existing mesh based navigation and motion synthesis 3) Furthermore while contact based optimization for human-scene interaction has indeed been used for mesh based representations, it has never ever been used in the context of neural rendering and we present the first effective use of such contact constraints for the 3DGS representation. We have updated the manuscript to include figures which were previously included in the supplementary material. The new figure 3 in the paper shows how contact optimization is necessary in 3dgs as without it the human penetrates through the chair for example.
>
> Here we also want to highlight that in the supplementary material - (Figure 1 and Figure 2 of supp mat) - we show what happens without our adaptations to path-finding and RL controllers  for navigation. Please note that all prior research on motion synthesis works in 3D mesh space - several algorithms exist to compute obstacle free paths in meshes but we for the first time show what adaptations are necessary to apply these algorithms to a scene represented using 3DGS.We also want to highlight the ablation studies presented in the supplementary material. There we use scannet++ scenes to evaluate our motion synthesis adaptations - as scannet++ provides both mesh and 3dgs scenes. We use 3dgs for synthesizing our motion but use the mesh for evaluation. We add the table from the supp mat here -
>
>
> | Method                     | Foot Contact (cm/s) $\downarrow$ | Penetration (m) $\downarrow$ | Jerk (m/s$^3$) $\downarrow$ | Goal Reaching (\%) $\uparrow$ |
> |---------------------------|-----------------------------------|-------------------------------|------------------------------|--------------------------------|
> | **Ours (full)**           | 0.89                              | 0.012                         | 0.47                         | 95.2                           |
> | w/o opacity culling       | 1.45                              | 0.042                         | 0.78                         | 42.6                           |
> | w/o walkability map       | 1.37                              | 0.021                         | 0.71                         | 84.5                           |
> | w/o optimization refinement | 1.11                            | 0.016                         | 0.55                         | 91.3                           |
>
> As is clear without our opacity culling and walkability adaptations, motion when synthesized in 3dgs scenes is not natural. We also want to highlight Figure4 provided in the supplementary. Our method works with monocular RGB videos of scenes - this is impossible with any existing mesh based algorithm. To test how existing mesh based baselines work for RGB videos - we reconstruct a mesh from RGB video using VGGT. Figure 4 of supplementary showcases the typical failure modes of this reconstruction - there are artifacts blocking paths - and hence the baselines which rely on meshes fail. We render these results using an Avatar+3DGS. These are visualized in supp video from 3:40. There are clear failures hence showcasing the necessity of our adaptations for 3DGS scenes.

---

> ### Author Response · Authors · 2025-11-17
>
> **2) Limitations**: We acknowledge that our method has limitations - indeed we have written about many of them in the conclusion.
>
> Computation: Over 5 motion sequences (on average 30 seconds) RL synthesis takes 10 seconds; optimization based synthesis takes 45 seconds (in total for transition in and out); contact optimization takes 60 seconds. In total for a 30 second sequence our whole algorithm takes approximately 300 seconds (including rendering and the forward pass through the StyleUnet network). We acknowledge this is not real time; and indeed we never claimed to have presented a real time method.
>
> We also acknowledge that our motion model is limited in that it mainly focuses on walking, sitting and performing some grabbing action  in 3D scenes - we have acknowledged this in the conclusion ourselves
>
> We also acknowledge that there are some cloudy artifacts - these are challenges inherent in using 3dgs as a representation because the final rendering for each pixel merges gaussian colors from humans and scenes and sometimes the contact is still unnatural despite our contact constraint.
>
>
> However despite this we want to highlight that this is the first ever study about using neural rendering - especially 3D gaussian splatting - for human animation in 3d scenes, and as such there is no real baseline for our problem (as is also acknowledged by *reviewer gJYF* ). While our method has limitations we believe it opens up a new direction of research at the intersection of human animation and neural rendering; and should serve as a catalyst for several ideas in the community.
>
> We also want to highlight, despite these limitations, that when we show naive raters our rendered results vs rendered results using a mesh based  baseline (baseline B) - timestep 4:34 in video - our results are considered more photoreal by a vast majority of users *(82.1 percent)* - (Table 2 of paper) hence showing that we have made progress towards photoreal animation in 3d environments
>
> We also want to highlight that we have presented the first ever method that works for synthesizing animation of humans in scenes reconstructed from monocular RGB videos - this is impossible with any existing mesh based algorithm. To test how existing mesh based baselines work for RGB videos - we reconstruct a mesh from RGB video using VGGT. Figure 4 of supplementary showcases the typical failure modes of this reconstruction - there are artifacts blocking paths - and hence the baselines which rely on meshes fail. We render these results using an Avatar+3DGS. These are visualized in supp video from 3:40. There are clear failures hence showcasing the necessity of our adaptations for 3DGS scenes.
>
>
> We thank the reviewer for their feedback. If there is anything else we can provide - clarification, new experiments, revision - for the reviewer to increase their rating - we would be happy to do so.

---

> > ### Comment · Reviewer_b3rV · 2025-11-25
> >
> > I appreciate the effort and clarifications the authors provided during the rebuttal. And I acknowledge that this work is the initial attempt to bring Gaussian avatars and 3D scenes together. However, I remain unconvined that the paper meets the standard for acceptance.
> >
> > After reading the rebuttals and reviews from other reviewers, I still beiieve that the main contribution of this paper lies at the system level rather than technical novelties --- while the methods include adaptations for Gaussian splatting, such changes are rather straightforward. At a system level, the limitations of the method, such as restricted interaction types and speed, constrain the method’s practical applicability so it is not ready for deployment in current AR/VR applications.

---

> ### Author Response · Authors · 2025-11-26
>
> We thank the reviewer for their feedback. We also thank them for acknowledging that “ this work is the initial attempt to bring Gaussian avatars and 3D scenes together”.
>
> **1. Adaptations for Gaussian splatting, such changes are rather straightforward** : We politely disagree. We believe (and indeed all other reviewers agree with us)  that our method is novel and our results are of interest to the wider community. If our idea were so trivial this would have been done before and as the reviewer themselves acknowledged this is the first ever method that tries to combine 3DGS avatars with 3DGS scenes. We again highlight that despite the limitations of our method all our experiments indicate we have made progress towards photoreal animation - our user study indicates that 82.1 percent of respondents consider our animation to be more photoreal than existing mesh based algorithms.
>
> **2.  Usage in real time AR/VR applications** : The reviewer is indeed correct that as it exists our method is not real time and deployable in real-time AR/VR applications but this is also true for a vast majority of current research on motion synthesis. For example Pinyoanuntapong et al. MaskControl - ICCV 2025 best paper finalist - uses the humanML3D dataset and after motion synthesis there is an inverse kinematics step which takes atleast 60 seconds to convert synthesized joint positions to SMPL parameters. While our method does indeed have limitations, we strongly believe (and all other reviewers agree with us) that our method will serve as a catalyst for further ideas in the research community and will serve as a strong baseline.

---

### Official Review · Reviewer_gJYF · 2025-11-01

**Soundness:** 1
**Presentation:** 2
**Contribution:** 3
**Rating:** 4
**Confidence:** 3

**Summary:**

The paper presents a method that enables virtual humans, represented using Gaussian splats, to navigate and interact in complex environments reconstructed as 3D Gaussian scenes. To achieve this, novel Gaussian-aligned motion synthesis and contact refinements are proposed.

**Strengths:**

As far as I know, this is the first work that models human-environment interactions in 3D Gaussian space. Although the interaction is quite basic (it cannot model intricate object manipulation), it is still impressive that the model can handle occlusion and collision. The video result quality is good and can inspire future works.

**Weaknesses:**

The critical weakness, in my opinion, is that there is no visual comparison (especially video comparison) with baselines. The authors attempted to design naive baselines (which I appreciate, since there were no previous 3D Gaussian works that enabled human-environment interaction), and the design is reasonable. However, they did not show any visual comparisons with them (i.e., baseline A and baseline B presented in the experiment section), and I think this is a significant weakness. Although readers might expect that the baselines would fail, to fairly validate the necessity of this work, the paper should have presented proper visual comparisons. I felt that the paper is a bit unpolished for submission. I would like to see other reviewers’ opinions on this and vote again. For now, I am leaning toward rejection and would encourage the authors to resubmit after including proper baseline visual comparisons and polishing the submission. Again, the results and the idea itself are very good.

**Questions:**

**Suggestions**

Authors don’t have to show this in the rebuttal, but for the revision, visual examples of handling collisions can be highlighted. My first impression was that this work simply places the human in the 3D scene.

Also, the overview figure does not emphasize the paper’s main contributions, such as contact refinement, and therefore is not very helpful for understanding the paper. I suggest adding the two main contributions in the overview figure (Gaussian-aligned motion synthesis and contact refinement).

---

> ### Author Response · Authors · 2025-11-17
>
> We thank the reviewer for complimenting the quality of our results. We are encouraged to see they recognize that we are presenting a novel idea about human motion synthesis in 3D environments:
>
> **1) Missing comparison (in supplementary video)**:  We invite the reviewer to watch the supplementary video at timestep 3:43 to 4:48 where we have already provided visual comparisons with baselines. This is almost towards the end of the video and easy to miss. We thank the reviewer for the suggestion regarding visual comparison in the main paper. We have updated the manuscript to also include images of comparisons of our method with these baselines.
>
> **2) Method figure**:  We thank the reviewer for this suggestion. We have updated the manuscript to update the method figure to emphasize what is novel. We hope the reviewer finds this new figure to be adequate. If the reviewer has further suggestions about the figure we would be happy to act upon those suggestions
>
> **3) Contact and occlusion**: We thank the reviewer for this suggestion. We had provided a figure in the supplementary which highlighted the effectiveness of our contact optimization; as per the reviewer’s suggestion we have now included this figure in the main paper instead of the supplementary.
>
> We are extremely encouraged by the appreciation the reviewer shows towards the quality of our results and the novelty of our idea. If there is anything else we can do, or actions the reviewer would like to see to further polish the manuscript, in order for them to increase their rating, we would be happy to act upon those suggestions

---

> > ### Comment · Reviewer_gJYF · 2025-11-17
> >
> > Dear authors, thank you for giving the pointer to the video comparison and incorporating my suggestions.
> >
> > I watched the whole video previously but misunderstood that the video was only showing the proposed method results, because in the comparison with baseline B, only baseline B was presented.
> >
> > For the comparison with baseline A, since it is an apple to apple comparison with the same scene and subject, I can see the clear improvement.
> >
> > For the comparison with baseline B, in the revised main paper, a different scene and a different subject were used. In the supplementary video, only baseline B was presented.
> >
> > Right now, it seems that both methods are able to make the subject sit on the sofa properly.
> >
> > Could you provide the apple to apple video comparison using the same scene and the same subject? I would like to decide after seeing the result.
> >
> >
> > (minor suggestion) Also, in the revised script, Line 466, there is a typo: plz → please.

---

> ### Author Response · Authors · 2025-11-17
>
> We thank the reviewer for the minor suggestion. We have updated the manuscript to correct this.
> For Baseline B our idea was not to provide an apple-to-apple comparison but provide a general idea as to what is possible with the best mesh based rendering for animation compared to the best 3DGS rendering with our animation algorithm.
> For Baseline B we aim to evaluate the best of mesh based rendering vs the best of 3DGS rendering. As such we use the highest quality 3D scenes, and the highest quality 3D human scans available as open source datasets - for the scene we use the Replica dataset, which to the best of our knowledge is the most photoreal mesh based scene dataset; and for humans we use a scan from the RenderPeople dataset - rigged to the SMPL body model.
> We compare renderings in this setup with the best of 3DGS renderings - using our method on SuperSplat scenes with AvatarX identities.
> To find the same scene+human combination for both our method and Baseline B would be difficult - because the data capture pipelines for 1) 3dgs vs mesh scene 2) 3dgs vs mesh human are fundamentally different. RenderPeople uses a rig of 250 DSLR cameras to capture a 3D human mesh - while the AvatarX dataset only uses 16 cameras and Behave dataset uses only 4. For 3DGS Avatar reconstruction motion of about 120 seconds inside the multiview camera setup is required while for a mesh capture only one frame is required. We could obtain a 3D human mesh from only 16 AvatarX cameras and rig it to the SMPL mesh but the quality would be much lower than RenderPeople scan.
> The way 3d scenes are reconstructed for replica (mesh) and SuperSplat (3dgs) are also fundamentally different - the replica reconstruction uses depth while some supersplat scenes use lidar. Additionally the replica original images are not available so we can’t reconstruct a 3DGS splat for Replica scenes of the same quality as the ones on SuperSplat.
> Based on this comparison (visualized at timestep 4:34) a vast majority of respondents in our user study (82.1 percent) consider our results to be more photoreal - showing that the best of 3DGS based rendering, with our motion algorithm, is preferred to the most photoreal mesh based rendering algorithm.
> We hope this provides further clarification regarding this issue; if there is anything further - experiments, clarifications -  the reviewer would like to see we would be happy to provide this. We thank the reviewer for the constructive feedback.

---

> ### Comment · Reviewer_gJYF · 2025-11-17
>
> Dear Authors,
>
> Thank you for the clarification, and now I understand the comparison setting difference.
> I have updated the score to accept. I think this is an interesting paper with good results. I vote for the acceptance of this paper :)
>
> Only suggestions I have is: after the rebuttal period, could you include more implementation details & insights and discussions to the supplementary material? As this is a very good direction that can enlighten the following works, it would be great if more details can be shared.

---

> > ### Author Response · Authors · 2025-11-17
> >
> > We are incredibly grateful to the reviewer for the comment and for all their feedback, which has improved the paper. If there is anything further - experiments, clarifications - we can provide for them to increase their rating further we would be happy to provide this. Thank you!

---

### Author Response · Authors · 2025-12-01
**To the ACs: Concerns addressed and two score upgrades during the discussion period**

Dear ACs


Thank you for your engagement in this reviewing process. This is a very unusual situation. Hence this summary statement will also be unusual. We highlight that all four reviewers acknowledge that we present the first ever method that animates 3DGS avatars in 3DGS scenes and are the first to demonstrate that neural rendering (with 3DGS) instead of classical rendering (with meshes) can be used as a functional component of human-scene animation pipelines. Two reviewers *gJYF* and *CKuQ* despite giving initial scores of 4 acknowledge in their initial reviews that our video results are “visually good”.


We also want to highlight two score upgrades that happened during the discussion period:


**gJYF**: The reviewer’s initial review was very positive. We quote *“it is still impressive that the model can handle occlusion and collision. The video result quality is good and can inspire future works”*  and *“Again, the results and the idea itself are very good.”* but the reviewer had concerns about there being no visual comparison with baselines - *“The critical weakness, in my opinion, is that there is no visual comparison (especially video comparison) with baselines”* and hence they gave an original score of 4. On Nov 17 in our response we wrote that though there is no visual comparison in the main paper we provide such comparisons in the supplementary video and that one particular comparison was structured slightly differently which perhaps made it difficult to understand that it's a comparison and not a result. We also included a new comparison figure in the revised manuscript. After these changes and clarifications, the reviewer *raised their score to 8 from 4 on Nov 17* and wrote *“Thank you for the clarification, and now I understand the comparison setting difference. I have updated the score to accept. I think this is an interesting paper with good results. I vote for the acceptance of this paper”*.


This exchange occurred on Nov 17 well before there was any public knowledge (approx Nov 27)  about leaks. We had proceeded with the discussion if it were a regular ICLR discussion. We hope the ACs agree that there is no evidence of any collusion or coercion in this exchange and the score was raised purely on grounds of our clarifications about the comparisons in the supplementary and the new figure.


**CKuQ**: The reviewer’s initial review was somewhat positive. We quote *“the work for the first time proposed to simultaneously solve the task of motion generation in the environment and photorealistic rendering”* and *“The approach demonstrates convincing qualitative results with sufficient visual realism”*. But the reviewer had several concerns: 1) baselines being complex systems 2) reliance on SMPL mesh and navigation mesh 3) what overall benefit does 3DGS have and a question about optimizing per-gaussian offsets and hence gave an initial rating of 4.  We addressed all these concerns in our reply on Nov 17 - i.e 1) baselines become complex because they need adaptations for 3DGS from mesh based systems 2) we do not rely on SMPL mesh and we only use SMPL pose (not mesh) and clarified that we use “navigation mesh” to mean “navigation region” 3) and pointed out several benefits of 3DGS. Upon these clarifications, the reviewer wrote that though they still have concerns *“Nevertheless, this work proposes to solve a novel task of photorealistic motion generation in the provided environment, and I believe it may be interesting to the community Therefore, I'm leaning towards raising the rating to Borderline accept”* on Nov 24 and *upgraded their score to 6 from 4*.


We again want to highlight that this occurred before there was any public knowledge about leaks. We hope the ACs agree that there is no evidence of any collusion or coercion in this exchange as well and the score was raised purely on merit of our arguments.


These score changes gave our paper **scores of 8,6,6,4 instead of 4,4,6,4 and an average score of 6 instead of 4.5** currently shown on openreview.


We still do not know who our reviewers are; we are firm believers in the peer-review system (despite its shortcomings) and would never tamper with the system in any way. We have put a lot of effort into the rebuttal. We would very very much hope that our discussions with the reviewers and the new scores *8,6,6,4* (before being reverted to their original scores) will be taken into consideration when taking the final decision about acceptance or rejection.

---

### Meta-Review · Area_Chair_61ku · 2026-01-05

**Summary:**

This paper uses 3D Gaussian Splatting to animate human avatars within 3D scenes. It successfully shows human-scene interactions like walking and sitting with high-quality rendering. However, the motion range is narrow and sometimes looks stiff. The experiment setting also lacks realistic lighting or shadows between the human and the scene. Before the rebuttal, reviewers questioned the novelty and the lack of visual baselines. The rebuttal fixed the baseline issues, but doubts about technical depth persist. Overall, the paper is below the acceptance bar and the authors are suggested to focus on more complex interactions for future work.

**Reviewer Concerns:**

The rebuttal addressed concerns about missing video comparisons and mesh usage. Yet, issues regarding slow runtime and limited interaction types remain. Reviewers also still feel the technical changes are too simple.

**Reviewer Scores:**

Reviewer gJYF would likely keep an accepted score as the rebuttal resolved their baseline concerns. Reviewer b3rV would remain at a reject score since they view the work as a simple system assembly. Reviewer CKuQ would probably move to a clear accept if the technical novelty were better highlighted. Reviewer 2xxi would keep their current positive score given the successful adaptation of reinforcement learning techniques.

---

### Decision · Program_Chairs · 2026-01-26

Reject